# Demand–Resource Profiles and Job Satisfaction in the Healthcare Sector: A Person-Centered Examination Using Bayesian Informative Hypothesis Testing

**DOI:** 10.3390/ijerph20020967

**Published:** 2023-01-05

**Authors:** Ivan Marzocchi, Valerio Ghezzi, Cristina Di Tecco, Matteo Ronchetti, Valeria Ciampa, Ilaria Olivo, Claudio Barbaranelli

**Affiliations:** 1Department of Psychology, Sapienza—University of Rome, 00185 Rome, Italy; 2Department of Occupational and Environmental Medicine, Epidemiology and Hygiene, Italian Workers’ Compensation Authority (INAIL), Monte Porzio Catone, 00078 Rome, Italy

**Keywords:** latent profile analysis, JD–R model, employee well-being, Bayesian informative hypotheses, healthcare sector

## Abstract

Work characteristics may independently and jointly affect well-being, so that whether job demands deplete or energize employees depends on the resources available in the job. However, contradictory results on their joint effects have emerged so far in the literature. We argue that these inconsistencies can be partially explained by two arguments in the contemporary literature in the field. First, most studies in the job design domain are based on classic variable-centered methodologies which, although informative, are not well suited to investigate complex patterns of interactions among multiple variables. Second, these studies have mainly focused on generic work characteristics (e.g., workload, control, support), and are lacking in occupational specificity. Thus, to overcome these limitations, in the current research we include generic and occupation-specific work characteristics and adopt a person-centered approach to (a) identify different patterns of interactions of job demands and resources in a sample of healthcare employees, and (b) determine the degree to which these patterns are associated with employee well-being. We involved a sample of 1513 Italian healthcare providers and collected data on key job demands (workload, emotional dissonance, patient demands and physical demands) and resources (control, management support and peers’ support). We focused on job satisfaction as a broad indicator of well-being. Latent profile analysis revealed four profiles of job demands and resources: high strain–isolated, resourceless, resourceful and active job on the ward. The results of Bayesian informative hypothesis testing showed the highest support for the hypothesis stating that healthcare employees belonging to the active job on the ward profile (medium–high demands, high resources) were the most satisfied. Conversely, employees belonging to the high strain–isolated profile (high demands, low resources) and the resourceless profile (medium–low demands, low resources) were the least satisfied. Overall, our study confirms the key role played by job resources in determining well-being in high-risk sectors, demonstrating that job satisfaction can develop both in challenging and less demanding situations. On a practical level, mapping the complexity of the healthcare psychosocial work environment has important implications, allowing for a better assessment process of employee well-being and helping to identify the most effective and fitting interventions.

## 1. Introduction

Stressful work characteristics, if not properly counterbalanced by appropriate resources, may result in undesirable consequences for employees, such as burnout and psychological distress [1]. Indeed, the job demand–resource (JD–R) model [2] states that the interplay among boundary conditions of job demands and resources may be of the greatest importance, over their individual effects, so that high job demands have a stronger negative impact on well-being if combined with limited job resources. Although convincing evidence has been provided for the unique effects of work characteristics on employee well-being [3], mixed findings have been found for their joint effects [4]. We contend that this limitation can be partially explained by two arguments of the contemporary literature in the field. First, most studies in the job design area have mainly focused on generic variables lacking in occupational specificity (e.g., workload, control, support). Second, classic variable-centered methodologies (e.g., regression analyses), although useful in providing insights into the association between different variables at a sample-wide level, are not well suited to investigate complex patterns of interactions among multiple work characteristics [5].

Drawing on the JD–R model [2], this study is aimed at (a) identifying different patterns of interactions of generic and contextual job demands and resources in a sample of healthcare employees, and (b) determining to what extent these patterns are associated with job satisfaction. We focus on the healthcare sector as it is one of the most at-risk work environments for employee well-being due to its inherent demanding work characteristics [6].

The main contributions of the current study are threefold. First, to overcome the aforementioned limitations, we adopt a person-centered approach using latent profile analysis (LPA), which provides a simultaneous examination of the interplay between different levels of job demands and resources. Second, we include a range of both generic and occupation-specific work characteristics which we believe better catch the peculiarities of the healthcare sector: workload, emotional dissonance, patient demands and physical demands as job demands, control, management support and peers’ support as job resources. These variables have proved to be critical for healthcare employees’ well-being in previous variable-centered studies (e.g., [7,8]). Third, due to the mixed evidence emerged so far, we use Bayesian informative hypothesis evaluation [9] to directly test several alternative hypotheses reflecting to what extent different profiles of work characteristics are associated with well-being.

### 1.1. Background: The JD–R Model and Job Satisfaction

Over the past few decades, several theoretical frameworks have been proposed to investigate the association between work characteristics and individual health. Among these, Demerouti and colleagues [10] have proposed the JD–R model, which classifies any work characteristic into two overarching categories: job demands and job resources. The model describes two distinct but related psychological processes to explain health issues (*health impairment process*) and positive implications for well-being (*motivational process*).

Job demands are the initiators of the health impairment process. They are described as those physical, psychological, social or organizational aspects of the work context associated with certain psychological and/or physiological costs [10]. Along with workload, which refers to the excessive intensity of job assignments that could determine health issues [10], we include in our investigation three healthcare-specific job demands: emotional dissonance, patient demands and physical demands. Emotional dissonance is the discrepancy between the experienced emotions and those that organizational contexts require to be displayed [11]. This job demand is critical in healthcare [12], and it was found to be associated with emotional exhaustion and disengagement [7,13]. Patient demands reflect the human side of clinical practice and describe the interactions perceived as stressful due to physical or psychological characteristics of the care recipients [14]. In their everyday working lives, healthcare employees are regularly confronted with patients who do not follow their advice, behave in hostile manners and make unrealistic requests [15]. Representing important sources of job dissatisfaction, poor mental health and burnout [16], negative interactions with patients are pivotal factors when examining healthcare workers’ well-being [17]. Finally, physical demands include repetitive movements, such as lifting, transferring and repositioning of patients, and the long-time adoption of inadequate and extreme postures. This job demand is crucial due to the many physical activities that healthcare employees are expected to perform [18]. Physical demands have been primarily associated with the emergency of musculoskeletal disorders [19], as well as negative health and low job satisfaction [20,21].

Conversely, job resources are the initiators of the motivational process. These are described as those physical, psychological, social or organizational aspects of the work context that can reduce job demands and their straining impact, are functional in achieving work goals, stimulate personal growth and, thus, determine well-being [22]. In the current study, we include control and social support as key job resources. We conceptualize job control as the employee’s autonomy to make decisions on the job and the breadth of skills used by the employee on the job. Next up, social support is the encouragement provided by both supervisors and colleagues to healthcare employees [23]. Control and social support have been associated with improved well-being in previous studies conducted in the healthcare sector (e.g., [24]).

With regard to employee health, in this study we focus on job satisfaction as a broad indicator of employee well-being and a potential correlate of the investigated job demands and resources. Defined as an individualized positive feeling and attitude toward a job [25], we included job satisfaction in our investigation as it is strongly linked with workers’ overall well-being [26] and, at the organization level, is associated with high performance and productivity [27,28], reduced absenteeism [29], lower turnover intention [30] and less counterproductive work behaviors [31]. With particular regard to the healthcare sector, previous studies have demonstrated the fundamental role of job satisfaction in this work context as being an important driver of quality of care, effectiveness, commitment to work and patient satisfaction (e.g., [32]). Conversely, job dissatisfaction has been associated with higher rates of medical errors and reduced patient safety (e.g., [33]).

### 1.2. The Interplay among Different Levels of Job Demands and Resources

A central tenet of the JD–R theory is that job demands and resources may interact and jointly affect well-being, so that whether job demands deplete or energize employees depends on the resources available in the job [2]. This assumption is consistent with the buffering hypothesis, which states that job resources may mitigate the negative impact of job demands on well-being, and the boosting hypothesis, which claims that high job demands may enhance the positive effects of job resources on well-being [2].

Consequently, the JD–R model assumes that a combination of high demands and low resources is more likely to determine poor employee well-being [10]. Next up, implying limited challenges and development opportunities, a combination of low demands and low resources is expected to be associated with poor well-being due to the boredom associated with low levels of job demands [23]. Because of the motivating role of job resources, a combination of low demands and high resources has been usually associated with moderate to high well-being [23]. Finally, a high demands–high resources configuration has been linked to moderate-to-high well-being [34]; consistent with the active learning hypothesis, high levels of resources may provide the necessary conditions for employees to evaluate medium-to-high levels of demands as motivating challenges which stimulate growth and achievement, thus promoting well-being [35,36].

Overall, the literature provides mixed evidence regarding the generalizability of the interactions as predicted by the JD–R model. On the one hand, several studies performed in the healthcare sector have found support for these interaction effects [37,38]. For example, Bakker and colleagues [39] found that the relationship between several job demands and emotional exhaustion disappeared when healthcare employees owned high levels of resources. Similarly, de Jonge et al. [18] showed that the association between job demands and job satisfaction was positive when job control was high, and was negative when the latter was low. Finally, Hakanen and colleagues [37] found that high job resources were more strongly associated with work engagement when job demands were high in a sample of Finnish dentists. On the other hand, a recent meta-analysis of the main and joint effects of job demands and resources found that neither the job demands–control nor job demands–social support interactions were significantly related to strain in almost all cases [40]. Moreover, a review performed by Taris [41] evidenced that only nine out of ninety tests performed provided support for these interaction effects. In line with previous research (e.g., [42,43]), we suggest that these inconsistent effects could be partially attributable to the limited specificity of the tools used to assess work characteristics and to the study methodology.

First, most of the studies and tools developed to investigate employee well-being are focused on generic work characteristics (e.g., [44,45]). Although work characteristics such as workload, control and social support are generally common in any workplace, there is a risk that these are not enough to extensively describe a specific psychosocial work environment and to capture contextual specificities [46]. Indeed, many researchers recommend the inclusion of occupation-specific work characteristics along with generic psychosocial risk factors in any occupational health investigation (e.g., [46,47]). This is consistent with the risk management paradigm [48], which advocates for the identification of all of the potential harmful work characteristics in workplaces to assess the risks to health and to set up corrective actions to reduce these risks. Moreover, the very same JD–R model is grounded on the assumption that every occupation is characterized by its own specific risk factors, which may not even be relevant in other work contexts [34].

Second, another aspect to pay attention to is the study methodology. For example, most studies in the job design domain are based on variable-centered approaches. Although informative, these methodologies are less appropriate in describing the simultaneous interplay among multiple variables, such as job demands and resources [5]. Hence, using different analytic approaches to investigate the joint influence of job demands and resources on well-being might represent a viable option [49]. A person-centered methodology may be particularly suitable for this purpose. Overall, while a considerable number of studies investigating the association between work characteristics and well-being have been performed, only a few of these have adopted a person-centered approach (e.g., [50,51,52]). Thus, to overcome these limitations, in the current study we focus on both generic (workload, control, support) and healthcare-specific characteristics (patient demands, physical demands, emotional dissonance). Moreover, we adopt latent profile analysis (LPA; for a recent overview see [53]), a model-based probabilistic clustering strategy which allows one to examine the unique constellations of generic and specific job demands and job resources experienced by healthcare employees.

### 1.3. JD–R Profiles in the Healthcare Sector

The healthcare sector is widely recognized as one of the most-at-risk work environments for employee well-being (e.g., [6,54]). Healthcare employees are exposed to a wide range of risk factors, such as quantitative and emotional demands, insufficient time to perform their job, adverse social behaviors and lack of resources (e.g., [55,56]). Moreover, the COVID-19 pandemic emergency has further intensified the pre-existing challenges of the sector [57]. Consequently, healthcare employees report the highest levels of work-related stress compared to other professionals [6], and experience high levels of burnout [58], psychological distress [59] and job dissatisfaction [60].

To the best of our knowledge, a very limited number of studies adopting a person-centered approach have been performed in the healthcare sector. For instance, Portoghese et al. [61] identified four latent profiles in a sample of healthcare professionals: isolated prisoner (high workload, low control and low support), participatory leader (low workload, high control and high support), moderate strain (average levels in all variables) and low strain (low workload, moderate levels of control and support). Among these, the isolated prisoner and moderate strain profiles showed the lowest levels of intrinsic work motivation. Investigating the interplay of four contextual job demands (stress due to residents, stress due to family members, stress due to working conditions and stress due to negative emotions) in a sample of nurses, Jenull and Wiedermann [62] revealed low-, moderate- and high-stress latent profiles, with the latter reporting the lowest job satisfaction. Bujacz et al. [63] found four distinct configurations in a cohort of Swedish nurses: supporting (low demands, high control and high support), demanding (high demands, low control and low support), constraining (low demands, low control and low support) and balanced (average levels in all variables). Members of the supporting profile had significantly lower levels of burnout than members of the other profiles, while members of the demanding profile experienced the highest burnout. In a study involving different professional groups, Holman [64] identified six different configurations using a two-step cluster analysis: active, saturated, team-based, passive–independent, insecure and high-strain. Further analyses revealed that healthcare roles, such as nurses and physicians, more likely belonged to the active (average-to-high levels of job demands and resources), saturated (high demands and high resources) and team-based profiles (moderate job demands and high resources). Job satisfaction was higher in the active group and lower in the high-strain group (characterized by high demands and low resources). Finally, Bujacz et al. [65] distinguished between four JD–R patterns in a sample of highly skilled Swedish workers: supporting (low demands and average-to-high resources), constraining (low demands and low resources), demanding (high demands and average-to-low resources) and challenging (high demands and high resources). Further analysis on professional groups revealed that the healthcare professionals were significantly more likely to be members of the demanding configuration. Moreover, this profile reported the lowest job satisfaction.

To summarize, different constellations of job demands and resources have emerged among healthcare employees. Therefore, our first research question (RQ) explores the number of homogeneous configurations of healthcare employees experiencing similar levels of job demands and resources:

**RQ1:** 
*How many latent profiles emerge in a sample of healthcare professionals when simultaneously considering both generic and occupation-specific job demands and resources?*


Although mixed evidence has been provided on the number of latent profiles to extract in this sample, in line with previous research which differentiates between passive, low-strain, high-strain and active jobs [52], we hypothesize to identify at least four latent profiles of healthcare employees covering all of the possible combinations of job demands and resources: (1) a high demands–low resources profile; (2) a high demands–high resources profile; (3) a low demands–high resources profile; and (4) a low demands–low resources profile.

### 1.4. Association between JD–R Profiles and Employee Well-Being

As depicted above, the results of previous studies regarding the simultaneous conjoint effect of job demands and resources on employee well-being have not always converged. Starting with the boosting hypothesis, there is no conclusive evidence that the highest level of employee well-being is especially associated with active jobs [66]. For instance, De Spiegelaere and colleagues [67] found that the configuration characterized by low demands and high resources reported higher levels of well-being than the profile characterized by high demands and high resources. In a similar vein, Moeller et al. [68] found those in the low demands–high resources configuration to be more engaged at work than those in the high demands–high resources profile. Landsbergis et al. [69] evidenced that the low demands–high resources profile had the lowest job dissatisfaction. Finally, Van den Broeck et al. [52] did not find significant differences between the high demands–high resources profile and the low demands–high resources profile in work engagement. These results suggest that adequate levels of employee well-being could also occur in more relaxed, less demanding jobs. Moreover, it is conceivable that even when many job resources are available, working under highly demanding conditions may be not only motivating, but also exhausting, especially in the long run [2]. With regard to the other JD–R configurations, one can also argue that if the highest well-being could occur in active jobs, then the lowest well-being could occur in passive jobs characterized by low demands and low resources [66].

Thus, through our RQ2 we intend to provide new evidence on the association between different JD–R profiles and employee well-being, represented by job satisfaction:

**RQ2:** 
*Are the emergent JD–R profiles differently associated with job satisfaction?*


Given the previous inconsistent results, we employ a Bayesian informative hypothesis testing approach to directly test to what extent different alternative hypotheses with inequality constraints are supported by the data [70]. According to Kluytmans and colleagues [9], there are several advantages to using informative hypotheses. For instance, they allow researchers to develop a pool of alternative hypotheses using their background knowledge, and to directly confront these hypotheses with empirical data. Conversely, the traditional null hypothesis significance testing (NHST) only allows for the testing of one’s expectations against the null hypothesis (i.e., the presence of an effect vs. no effect). Moreover, the use of informative hypotheses largely eliminates the multiple testing problem that occurs when researchers need follow-up tests to explore an omnibus effect (i.e., the Bonferroni correction for multiple comparisons).

Guided by the JD–R model’s theoretical underpinnings, we hypothesize that an eventual high demands–high resources profile would report the highest job satisfaction, followed by the low demands–high resources profile. Conversely, we hypothesize that eventual low demands–low resources and high demands–low resources profiles would report the lowest job satisfaction.

## 2. Materials and Methods

### 2.1. Procedure and Participants

This study is based on data collected in 2018 by the Italian Workers’ Compensation Authority (INAIL) in two Italian public hospitals. INAIL’s aim was to conduct an explorative study to test several integrative tools which reflected the specificities of the healthcare sector. Thus, the Italian methodology to tackle work-related stress risk in the healthcare sector was applied [71,72]. INAIL’s methodology was developed to enable Italian organizations to comply with the national legal requirement to assess and manage psychosocial risks along with other risks for health and safety in the workplace. With the aims of effectively managing the psychosocial risks at work and improving workers’ health and well-being, some integrative tools were developed specifically for healthcare needs and were published by INAIL in 2022. In the current study, only the data gained from employees as part of the in-depth assessment phase were analyzed. During this phase, a questionnaire was distributed to 6687 employees from the two public hospitals that agreed to get involved in INAIL’s exploratory study. The first page of the questionnaire provided information about the collection and use of personal data in the research, in line with the General Data Protection Regulation (GDPR, Regulation n° 2016/679). All participants marked their informed consent on that page before they continued with the compilation. The confidentiality and the anonymity of the answers were emphasized. A total of 1905 questionnaires were collected; of these, 1513 were from healthcare providers (e.g., physicians, nurses), while the remaining 392 were from administrative workers. To focus on similar roles and capture as much as possible the shared job peculiarities and potential risk factors, in this study we decided to only include the healthcare providers. More than two thirds of participants (73.4%) were females, and 26.6% were men, mirroring the proportions of the healthcare workforce in Italy where women represent around 70% of the total. The majority were aged between 31 and 50 (51.8%), and almost the entire sample was composed of Italian workers (98.7%). With regard to work contracts, 93.2% of the workers had a permanent contract, followed by fixed-term (4.5%) and interim contracts (1.2%). Most of the participants (60.4%) were employed in shift work, and 59.9% of them worked on both day and night shifts. Finally, in terms of average job tenure, participants had been working in the same unit for 10.4 years (SD = 9.4), and in the same company for 16.7 years (SD = 11.4).

### 2.2. Measures

#### 2.2.1. Job Demands

##### Workload

Workload was assessed through three items from the subscale Demands of the Italian version of the Management Standards Indicator Tool (MS-IT; [73]). A sample item was, “I have unachievable deadlines”. In accordance with the original measure (Edwards et al. 2008), employees were asked to answer questions using a five-point Likert scale (1 = strongly disagree to 5 = strongly agree). Cronbach’s α was 0.78 and McDonald’s ω was 0.79.

##### Patient Demands

Patient demands were investigated through a tailored version of the Dormann and Zapf’s scale [74] used to measure service workers’ perception of highly demanding customers (seven items; sample item: “Some patients always demand special treatment”). For this variable, the participants’ answers were recorded on a five-point Likert scale (1 = strongly disagree to 5 = strongly agree). Cronbach’s α was 0.85 and McDonald’s ω was 0.86.

##### Emotional Dissonance

Emotional dissonance was assessed through a three-item scale developed by Zapf and Holz [75], successively tailored to healthcare professionals by Consiglio [76]. A sample item was, “At work it happens that I cannot express the emotions I feel”. Employees were asked to answer questions using a five-point Likert scale (1 = strongly disagree to 5 = strongly agree). Cronbach’s α was 0.68 and McDonald’s ω was 0.67.

##### Physical Demands

Physical demands were assessed through a three-item scale to investigate ergonomic risk developed for the purpose of the study. A sample item was, “I have to lift, push or pull heavy loads (including patients)”. For this variable, the participants’ answers were recorded on a five-point Likert scale (1 = strongly disagree to 5 = strongly agree). Cronbach’s α was 0.70 and McDonald’s ω was 0.74.

#### 2.2.2. Job Resources

##### Control

Control was assessed through three items from the MS-IT [73], reflecting the autonomy that employees have in exercising their own work activities. A sample item was, “I have a choice in deciding how I do my work”. Employees were asked to answer questions using a five-point Likert scale (1 = strongly disagree to 5 = strongly agree). Cronbach’s α was 0.82 and McDonald’s ω was 0.80.

##### Management Support

Management support was assessed through three items from the MS-IT [73], which focus on the encouragement and support provided by employers and management in regard to employees. A sample item was, “My line manager encourages me at work”. Employees were asked to answer questions using a five-point Likert scale (1 = strongly disagree to 5 = strongly agree). Cronbach’s α was 0.90 and McDonald’s ω was 0.90.

##### Peers’ Support

Colleagues’ support was assessed through three items from the MS-IT [73]. A sample item was, “I get help and support I need from colleagues”. Employees were asked to answer questions using a five-point Likert scale (1 = strongly disagree to 5 = strongly agree). Cronbach’s α was 0.86 and McDonald’s ω was 0.85.

#### 2.2.3. Employee Well-Being

##### Job Satisfaction

Job satisfaction was assessed through a one-item measure of general satisfaction with job, (“Generally speaking, I’m very satisfied with my job”). Statements were answered on a five-point Likert scale (1 = strongly disagree, 5 = strongly agree).

### 2.3. Data Analysis

As a preliminary analysis, we tested the psychometric properties of the tools used to measure the study variables. In line with guidelines provided by Morin et al. [77], we contrasted confirmatory factor analysis (CFA) and exploratory structural equation modeling (ESEM) models to compare alternative factor structures of job demands and resource scales. ESEM is a method that incorporates the advantages of the less restrictive exploratory factor analysis (EFA) and the more advanced CFA. Recent studies have shown that ESEM usually results in an improved model fit as well as deflated inter-factor correlations, providing a more realistic representation of the data [78]. The ESEM model was specified in a confirmatory manner using the orthogonal target rotation, which allows cross-loadings across items, but constrains them to be as close to zero as possible [79]. We evaluated model fit inspecting the following indices: comparative fit index (CFI), Tucker and Lewis index (TLI), root mean square error of approximation (RMSEA) and its 90% confidence interval (90% CI) and standardized root mean square residual (SRMR). We also report the chi-square test of model fit, which provides an indication of the difference between the observed covariance matrix and the model covariance matrix. We adopted the following criteria to assess model fit: CFI ≥ 0.90, TLI ≥ 0.90, RMSEA ≤ 0.08, SRMR ≤ 0.08 [80]. Moreover, the chi-square difference test was used to statistically compare the fit of the two models. Factor scores, which represented the manifest indicators of the LPA models, were calculated and stored from the best fitting measurement model. Different to observed means, factor scores preserve the underlying nature of the latent constructs and partially control measurement errors by giving less weight to items with lower factor loadings [81].

To answer RQ1, we applied LPA to identify latent subgroups of workers sharing similar perceptions of the seven work characteristics. We estimated LPA solutions considering one to eight latent profiles using M*plus* 8.6 and robust maximum likelihood estimators (MLR; [82]). The number of initial stage random starts was set at 10,000, with the 500 best solutions retained for the final stage of the optimizations. The number of iterations was set at 1000. We relied on several criteria to decide upon the number of profiles to retain. First, we inspected the following fit indices: Akaike’s information criterion (AIC), Bayesian information criterion (BIC), sample-size-adjusted Bayesian information criterion (SABIC), the approximate weight of evidence criterion (AWE), the Lo–Mendell–Rubin test (LRT) and the adjusted LRT (Adj-LRT). Although absolute criteria for evaluating model fit are yet to be established, lower AIC, BIC, SABIC and AWE are usually preferred. The LRT and Adj-LRT provide *p* values assessing whether adding a class leads to a statistically significant improvement in model fit; a non-significant *p* value for a *k* class solution provides support for the *k* − 1 latent profile solution. Moreover, we inspected the relative entropy of the best fitting model, a measure of overall classification applied to profiles which gives an indication of the degree of distinctiveness between latent classes. Relative entropy can be considered sufficient when its value exceeds 0.70 [83]. Finally, we also considered parsimony and meaningfulness to decide upon the number of profiles to extract [84]. Indeed, when a subgroup includes less than 5% of samples, the latent profile may not be meaningful and removal of the group should be considered for the sake of model parsimony [85]. After having identified the optimal LPA solution, we performed a chi-square test and one-way analysis of variance (ANOVA) with Bonferroni post hoc pairwise comparisons to assess the distinctiveness between the profiles generated by the LPA with regard to sociodemographic aspects and the job demands and resources used for the clustering procedure. We used the software IBM SPSS 23 for this specific analysis.

To answer RQ2, we tested and compared several competing informative hypotheses aligned with the previous literature using the software JASP—version 0.15 [86] and the Bayesian informative hypothesis evaluation software (BAIN; [87]) implemented in JASP. For this set of analyses, the statistics of interest were the Bayes factor (BF) and the posterior model probability (PMP). The BF quantifies the evidence of each informative hypothesis with respect to an unconstrained hypothesis (i.e., a hypothesis that presumes no specific ordering of the variables). A BF > 20 can be interpreted as a sign of strong support from the data to the informative hypothesis [70]. We also computed the relative Bayes factor (BF_x,y_), a ratio that quantifies how much the data provide support to hypothesis X when compared to hypothesis Y; if this ratio is >1, one can conclude that hypothesis X should be preferred over hypothesis Y. The PMP is a standardized version of a single BF divided by the sum of all BFs, and it expresses support in the data for the hypothesis at hand given the set of all the other hypotheses under evaluation. PMPs have values between 0 and 1 and sum to 1 for the hypotheses in the set under consideration. Finally, we computed Cohen’s *d* as the effect size of the differences between the latent profiles with regard to their mean values of job satisfaction.

## 3. Results

### 3.1. Descriptive Statistics Results

Mean scores for the study variables ranged from 2.4 to 3.8 (Table 1), with most of them around the mid-point. Except for workload (kurtosis = −1.16), all the scales exhibited non-problematic levels of skewness and kurtosis [88]. However, Mardia’s multivariate test suggests that multivariate normality was not reached (Mardia’s multivariate skewness coefficient = 4.55; χ^2^ = 1004.98; *p* < 0.001; Mardia’s multivariate kurtosis coefficient = 86.62; z = 13.84; *p* < 0.001). Hence, we performed CFA and ESEM using M*plus* 8.6 and the MLR estimator, as they correct the standard error of model parameters when there are slight departures from univariate and multivariate normal distributions of the data [82].

### 3.2. Psychometric Characteristics of the Tools

CFA results showed a non-satisfactory fit of the model: MLRχ^2^ _(*df* = 254)_  =  1555.29, *p*  <  0.001; CFI = 0.89; TLI = 0.87; RMSEA = 0.062 (90% CI: 0.059–0.065); SRMR = 0.058. On the contrary, the same examination applied to ESEM yielded a good fit for the model: MLRχ^2^ _(*df* = 146)_  =  585.26, *p*  <  0.001; CFI = 0.96; TLI = 0.93; RMSEA = 0.048 (90% CI: 0.044–0.052; *p* = 0.82); SRMR = 0.018. Moreover, the Satorra–Bentler scaled chi-square difference was significant (SBΔχ^2^
_(Δ*df* = 108)_ = 963.12, *p* < 0.001), suggesting that the CFA model fitted the data significantly worse than the ESEM model. Thus, we retained the ESEM solution to derive factor scores to use in the LPA. All the item factor loadings in the target factors were higher than loadings in the others. Except for one item from the physical demands scale showing low target factor loadings (λ = 0.32), the interpretation of factor loadings of the seven factors revealed that each scale was well defined by its items (ranging from λ = 0.42 to λ = 0.89; see Appendix A).

Table 1 shows the intercorrelations between job demands, job resources and job satisfaction.

### 3.3. Latent Profile Analysis Results

Since the AIC, CAIC, BIC, SABIC and AWE indices decreased as additional profiles were added (Table 2), we relied instead on a graphic examination (“elbow-plot”). The elbow plot showed an evident flattening of the slope around two profiles, and a slight flattening around four profiles and six profiles (Figure 1). The indicators of parsimony (LRT and AdjLRT) supported the models with two (*p* < 0.001), four (*p* < 0.05), six (*p* < 0.001) and seven profiles (*p* < 0.01). In the solution of Profile 7, the smallest profile included only 5% of the participants; thus, it was rejected. When we compared the solution of Profile 4 with the solution of Profile 6, we found that the two additional profiles did not provide meaningful additional information, being very similar in shape to the other two profiles. Moreover, in Solution 6, a profile emerged with 5.6% of the sample, a percentage that was only slightly acceptable. Consequently, we chose the more parsimonious solution of Profile 4. Relative entropy for this solution was adequate (>0.70).

### 3.4. Characteristics of the JD–R Profiles

As Figure 2 and Table 3 show, healthcare employees belonging to Profile 1 (N = 118; 8.9%) were characterized by mixed levels of job demands (high workload, but low emotional dissonance; very low physical and patient demands), and very low control and support resources; thus, we labeled this profile *resourceless* to reflect the very limited amount of resources reported by these employees. Profile 2 (N = 287; 21.7%) was characterized by low workload, low emotional dissonance, low physical and patient demands and high control and support; consequently, we labeled this profile *resourceful*. Healthcare employees belonging to Profile 3 were the majority (N = 524; 39.6%), and reported high workload, high emotional dissonance, high patient and physical demands and low control and support; hence, we renamed this profile *high strain–isolated*. Finally, Profile 4 (N = 395; 29.8%) was characterized by low workload and low emotional dissonance, very high physical and patient demands and high control and support; thus, we labeled it *active job on the ward* to reflect the high interaction with patients and the high support provided to these workers by colleagues, managers and the overall organization (through job control).

Thus, answering RQ1, we identified four different latent configurations to represent the complex interactive patterns among general and specific job demands and resources in the considered sample. Basic descriptive statistics for each profile are reported in Table 4. Compared to the other profiles, the *resourceless* profile was characterized by a lower proportion of females. Regarding age, the *resourceful* profile was characterized by a higher proportion of employees who were more than 50 years old and a lower proportion of employees between 31 and 50 years old; conversely, the *active job on the ward* profile was characterized by a higher proportion of employees who were less than 30 years old and between 31 and 50 years old, and a lower proportion of employees who were more than 50 years old. The profiles did not differ in terms of work contract. The *resourceful* profile was characterized by a lower proportion of employees performing shift work; conversely, the *high strain–isolated* and the *active job on the ward* profiles were characterized by a higher proportion of employees performing shift work. Finally, the *resourceful* profile was characterized by the highest organizational tenure.

### 3.5. Association between JD–R Profiles and Job Satisfaction

We quantified the association between the JD–R profiles and job satisfaction using Bayesian informative hypothesis testing. First, we defined an informative hypothesis which assumed that the different constellations of job demands and resources in healthcare employees did not produce any differences in job satisfaction (H_0_). This is the Bayesian version of the classical null hypothesis of one-way ANOVA models evaluated using omnibus tests within the NHST approach [70], and it served as a benchmark for the other substantive competitive hypotheses. Thus, we posited that:H_0_: µActive = µResful = µResless = µStrain (1)
where µ represents the average level of job satisfaction within the four given latent profiles: *active job on the ward* (“Active”), *resourceful* (“Resful”), *resourceless* (“Resless”) and *high strain–isolated* (“Strain”).

Our first informative hypothesis reflected the JD–R assumptions and posited that the “Active” group was the most satisfied, followed by the “Resful” group, the “Resless” group (which partially corresponded to a passive job profile) and the “Strain” group:H_1_: µActive *>* µResful > µResless *>* µStrain (2)

The next set of informative hypotheses reflected the mixed evidence illustrated in the previous theoretical sections. Starting from the two “positive” configurations (*active job on the ward*, *resourceful*), two of the informative hypotheses assumed that the group characterized by medium–high job demands and high job resources would have reported the highest job satisfaction (H_3_, H_4_; e.g., [64]). Next up, a set of three hypotheses underlined the prominent role played by the low demands—high resources configuration in affecting satisfaction at work the most (H_2_, H_6_, H_9_; e.g., [67]). Moreover, following the findings of Van den Broeck and colleagues [52], we drafted three hypotheses in which we assumed there would be no differences in job satisfaction between the *active job on the ward* and the *resourceful* groups (H_5_, H_7_, H_9_). Switching to the two “negative” configurations (*resourceless*, *high strain–isolated*), a first set of hypotheses stated that the high demands–low resources group would have reported the lowest job satisfaction (H_2_, H_7_; [34,64]). Conversely, in line with the reflections of Taris et al. [66], who argued that if well-being occurs especially in active jobs then the lowest well-being could occur in passive jobs, several hypotheses assumed that the *resourceless* group would have reported the lowest job satisfaction (H_3_, H_6_, H_8_). However, as our *resourceless* group did not match totally with the prototypical “passive” job profile, two informative hypotheses assumed no differences in job satisfaction among the *resourceless* and the *high strain–isolated* groups (H_4_, H_9_).
H_2_: µResful *>* µActive > µResless *>* µStrain (3)
H_3_: µActive > µResful > µStrain > µResless (4)
H_4_: µActive > µResful > µResless = µStrain (5)
H_5_: µActive = µResful > µStrain = µResless (6)
H_6_: µResful > µActive > µStrain > µResless (7)
H_7_: µActive = µResful > µResless > µStrain (8)
H_8_: µActive = µResful > µStrain > µResless (9)
H_9_: µResful > µActive > µResless = µStrain (10)

Table 5 reports results from the Bayesian evaluation of the competitive informative hypotheses. The two informative hypotheses that were more supported by the data were H_4_ and H_1_. Indeed, BFs and PMPs were higher for H_4_ (BF_4_ = 27.43; PMP = 0.53) and H_1_ (BF_1_ = 23.08; PMP = 0.45). In addition, BF_4,1_ suggested that H_4_ was 1.19 times more likely than H_1_ to occur. Thus, although the support of H_4_ with respect to H_1_ is far from strong, we can conclude that H_4_ is the most likely informative hypothesis among those tested in our study.

Table 6 reports the effect sizes related to latent profiles’ differences in job satisfaction. As expected, Cohen’s *d* was significant in all cases, except for the difference between Profile 1 (*resourceless*) and Profile 3 (*high strain–isolated*) (*d* = 0.17, *p* = 0.61). The other differences were medium-to-low (*resourceless* versus *resourceful*: *d* = −0.34, *p* = 0.01; *resourceful* versus *active job on the ward*: *d* = −0.33, *p* < 0.001), medium (*resourceful* versus *high strain–isolated*: *d* = 0.51, *p* < 0.001), medium-to-high (*resourceless* versus *active job on the ward*: *d* = −0.68, *p* < 0.001) and high (*high strain–isolated* versus *active job on the ward*: *d* = −0.84, *p* < 0.001) (Cohen, 1992).

Thus, answering RQ2, our data most supported the hypothesis stating that employees belonging to the *active job on the ward* configuration were the most satisfied (*M* = 4.3, *SD* = 0.8). This was followed by the *resourceful* profile (*M* = 3.9, *SD* = 0.8). Although the *high strain–isolated* configuration reported the lowest mean value of job satisfaction (*M* = 3.5, *SD* = 1.1), this value was not different from that obtained by the *resourceless* profile (*M* = 3.6, *SD* = 1.1).

## 4. Discussion

In the present study, we aimed (a) to identify different profiles of healthcare employees experiencing similar levels of generic and contextual job demands and resources, and (b) to elucidate their association with job satisfaction as an indicator of employee well-being. To the best of our knowledge, a very limited number of studies have been previously performed using a similar methodology. Moreover, even less have seen the involvement of healthcare employees, which, according to recent and not-so-recent research, are exposed to a wide range of risk factors and suffer from a broad spectrum of health issues (e.g., [6,54,57,59]). Overall, our study confirms that unique constellations of job demands and resources can emerge in this sector, and that these are differently associated with job satisfaction.

Through LPA [53], we identified four JD–R configurations: a high demands–low resources profile (*high strain–isolated*); a low demands–high resources profile (*resourceful*); a mixed demands–high resources profile (*active job on the ward*) and a mixed demands–very low resources profile (*resourceless*). The first three profiles largely correspond to the job types already proposed within the job design domain [34]. Instead of a “passive” job profile combining low levels of demands and resources, we found a mixed demands–very low resources profile (*resourceless*). This result is only partially surprising, since the low demands–low resources constellation had not already emerged in previous person-centered examinations among healthcare employees (e.g., [61,63]). Moreover, Holman [64] found that the proportion of healthcare employees belonging to a passive job profile was relatively low compared to other professionals. The absence of a passive profile, coupled with the fact that most healthcare employees belonged to the *high strain–isolated profile* (39.6%), may be interpreted as cues of the highly demanding work conditions that these professionals are often subjected to [6].

We used Bayesian informative hypothesis testing [70] to investigate the association between the different latent profiles and job satisfaction. Overall, our results confirm almost totally the JD–R model assumptions. We found that the *active job on the ward* profile reported the highest level of job satisfaction, followed by the *resourceful* profile. These results corroborate the boosting hypothesis, which claims that job resources become more salient and gain their full motivational potential when employees are confronted with challenging levels of job demands [34]. Notably, the boosting hypothesis has not always found support in previous person-centered studies. However, in our research we included several contextual work characteristics which helped us in describing the healthcare psychosocial work environment in a more nuanced way. Indeed, our *active job on the ward* profile is characterized by a lower-than-average workload and emotional dissonance, but high levels in two occupation-specific demands (interaction with patients and physical demands) and high control and social support. Both interaction with patients and physical demands have been linked to negative outcomes in previous studies performed in the healthcare sector (e.g., [21]). However, helping others has been linked to a sense of well-being rather than burnout or fatigue when managed properly [89,90]. Our results suggest that this is true when adequate levels of job resources are afforded to employees. We can argue that the *active job on the ward* configuration may represent an adequate match between demands and resources for healthcare employees’ well-being.

The importance of job resources in promoting well-being is further confirmed by the high job satisfaction reported by the *resourceful* group. This configuration has been associated with a number of salutogenic outcomes, such as work engagement, in prior person-centered research (e.g., [67]). Our study corroborates the main effects of job resources on well-being, confirming that high levels of job satisfaction can also develop in more relaxing and less demanding work situations.

The *high strain–isolated* profile, characterized by high demands and low resources, obtained the lowest job satisfaction. This result supports another important assumption of the JD–R model, namely that ill-being develops when job demands are excessive and job resources are limited [10]. Moreover, this result is consistent with the *iso–strain hypothesis* of the demand–control–support model [23], which claims that especially the synergic combination of high demands, low control and low social support may result in detrimental consequences for employee well-being. Notably, we did not find significant differences between the *high strain–isolated* and the *resourceless* profiles. We could hint that the *resourceless* profile is a cross between the prototypical *passive* and *high strain–isolated* profiles [23], as it consists of high workload, very low healthcare-specific job demands and very low resources. Thus, a low level of job satisfaction reported by such a configuration is hardly surprising.

### 4.1. Implications for Practice in the COVID-19 Era

The COVID-19 pandemic emergency has had detrimental consequences on the way employees work within organizations [57]. Although all the data were collected before the start of the pandemic, we believe our results could provide practical indications which are still valid in these troubled times.

First, our results shed light on the importance of increasing job resources to improve well-being in the healthcare sector, perhaps to a greater extent than reducing job demands (if they are not excessive). Job resources appear to be particularly critical not only due to their intrinsic motivating potential, but also because they allow one to effectively deal with job demands that cannot be eliminated or reduced [91]. In the COVID-19 era, providing healthcare employees with a vast amount of job resources, whether they are material (i.e., personal protection equipment), social (i.e., improved supervisor support) or organizational (i.e., flexible work options, whenever possible), is paramount to reducing the negative impact of traditional and emerging job demands, and thus to protecting employee well-being. In this vein, our study further highlights the essential roles played by control and social support for employee well-being. Therefore, healthcare organizations should invest as much as possible in the autonomy of employees by allowing them to determine the pace, sequence and methods when performing their job activities. Moreover, actions aimed at strengthening management support, such as improving managers’ abilities in giving feedback, enhancing communication and adopting adequate leadership styles, may be used as part of an occupational health intervention in this sector. In turn, these actions may enable employees to work together effectively and collaboratively, thus increasing team cohesion and colleagues’ support.

Second, we highlight the importance of targeting both generic and contextual job demands and resources in any strategy for the assessment and management of the psychosocial work environment, particularly in high-risk sectors such as the healthcare one [92]. This approach is consistent with the risk management paradigm [48], which calls for the identification of all of the potential harmful work characteristics in workplaces to assess the risks to health and to design fitting interventions aimed at reducing such risks. Nowadays, a careful analysis of all the potential risks for the health of workers seems necessary, since COVID-19 has not only exacerbated some pre-existing stressful working conditions, but has also determined the emergence of new contextual stressors [93].

Third, mapping the complexity of the healthcare psychosocial work environment into several distinct patterns may simplify any assessment process concerning poor employee well-being. Recent studies have highlighted that the European organizations are more likely inclined to implement secondary- and tertiary-level interventions in their action plans [94,95]. Focusing less on job demand reduction and more on increasing job resources is perceived as less complex and time-consuming, and this does not require specific competencies in work design and organization [96,97]. Nevertheless, interventions put in place are often ineffective and do not seem to achieve the desired outcomes [98]. Our findings may allow one to quickly identify the subpopulations of healthcare employees whose well-beings are more at risk, and to develop effective and targeted job re-design interventions. For example, the *high strain–isolated* profile would especially benefit from organizational interventions aimed at simultaneously reducing job demands and increasing job resources, while the *resourceless* profile would benefit from actions aimed at improving job resources. At the same time, employees in the *resourceful* and the *active job on the ward* profiles would benefit from interventions to further boost, or at least maintain, their access to job resources.

### 4.2. Study Limitations and Future Research

Our study is not exempt from limitations. First, we collected all information using self-reported measures; thus, it is possible that common method bias and social desirability may have affected our results [99]. Future studies should include objective measures from different sources for job demands and resources (e.g., actual number of working hours) and outcomes (e.g., turnover rates, register-based sickness absences) to overcome this limitation.

Second, being cross-sectional in nature, the present study only allows one to examine the concurrent differences in job satisfaction among the JD–R profiles that emerged. Longitudinal and diary studies adopting lagged or intensive time intervals should be performed in the future to investigate the causality and stability/changeability of these characteristics.

Third, we relied on a convenience sample of Italian healthcare workers. Further investigations would be beneficial to investigate whether our findings could be replicated using a different sample of healthcare employees and in another cultural context.

Finally, our well-being investigation is limited to job satisfaction, which is assessed using a single-item measure. Previous studies have demonstrated that job satisfaction is strongly associated with workers’ overall well-being [26] and, in the healthcare sector, with improved quality of care, commitment to work, patient satisfaction and reduced medical errors (e.g., [32,33]). Notwithstanding this, future research may benefit from the inclusion of additional pathogenic (e.g., burnout, psychological distress) or salutogenic outcomes (e.g., work engagement, commitment) to investigate whether similar results could be obtained. Moreover, although prior research has supported the acceptability of single-item overall job satisfaction metrics as psychometrically sound instruments [25], future studies may use validated instruments to better explore the association between different JD–R profiles and various facets of job satisfaction.

## 5. Conclusions

The current study extends the existing literature on well-being in the healthcare sector by complementing the variable-centered methodologies with a person-centered approach. LPA revealed four different configurations of job demands and resources. Using Bayesian informative hypothesis testing, we found that the employees belonging to the *active job on the ward* profile were the most satisfied, while the employees belonging to the *high strain–isolated* and the *resourceless* profiles were the least satisfied. Our results further underline the paramount role played by job resources in employee well-being. Thus, organizational interventions intended to increase job resources and to reduce excessive job demands would be particularly beneficial for employees and organizations, especially in highly demanding work environments such as the healthcare sector.

## Figures and Tables

**Figure 1 ijerph-20-00967-f001:**
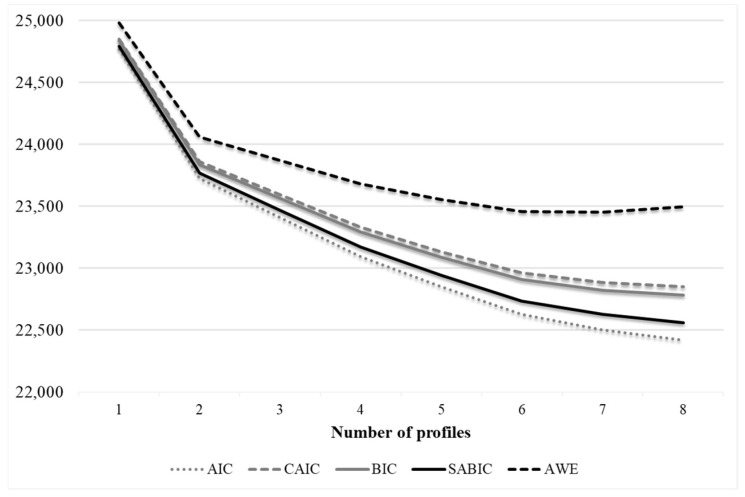
Elbow plot for the information criteria. Note. The number of profiles extracted in an LPA model are on the X-axis and the fit statistics are on the Y-axis.

**Figure 2 ijerph-20-00967-f002:**
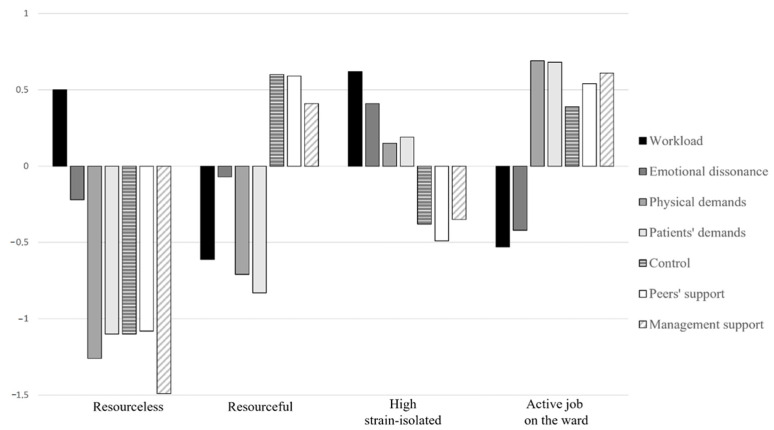
Latent profile of job demands and resources.

**Table 1 ijerph-20-00967-t001:** Descriptive statistics for the study variables and ESEM factor score correlations.

Variable	Mean (SD)	Skew	Kurt	1.	2.	3.	4.	5.	6.	7.	8.
1. Workload	2.4 (0.8)	0.43	−1.16	(0.78)							
2. Emotional dissonance	3.2 (0.8)	−0.25	−0.23	0.45 ***	(0.68)						
3. Patient demands	3.3 (0.8)	−0.33	−0.11	0.06 *	−0.02	(0.85)					
4. Physical demands	3.0 (1.0)	−0.10	−0.80	−0.07 **	0.11 ***	0.57 ***	(0.70)				
5. Control	3.5 (0.9)	−0.50	−0.02	−0.42 ***	−0.10 ***	0.14 ***	0.04	(0.82)			
6. Peers’ support	3.7 (0.8)	−0.70	0.59	−0.39 ***	−0.20 ***	0.02	0.14 ***	0.48 ***	(0.86)		
7. Management support	3.5 (1.1)	−0.55	−0.45	−0.36 ***	−0.10 ***	0.18 ***	0.29 ***	0.39 ***	0.54 ***	(0.90)	
8. Job satisfaction	3.8 (1.0)	−0.73	−0.01	−0.25 ***	−0.40 ***	0.07 *	0.02	0.26 ***	0.32 ***	0.33 ***	-

Note. SD = standard deviation; skew = skewness; kurt = kurtosis; Cronbach’s alpha values are reported in brackets. *** *p* < 0.001; ** *p* < 0.01; * *p* < 0.05.

**Table 2 ijerph-20-00967-t002:** Fit indices for the eight estimated solutions of job demand and resource profiles.

Number of Profiles	AIC	CAIC	BIC	SABIC	AWE	LRT (*p*)	AdjLRT (*p*)	Entropy	Smallest Profile (%)
1	24,766	24,853	24,839	24,794	24,982	-	-	-	-
2	23,724	23,860	23,838	23,768	24,062	<0.001	<0.001	0.722	35.3
3	23,409	23,594	23,564	23,469	23,870	>0.05	>0.05	0.703	14.4
4	23,095	23,330	23,292	23,171	23,679	<0.05	<0.05	0.729	8.9
5	22,849	23,134	23,088	22,942	23,556	>0.05	>0.05	0.774	7.1
6	22,627	22,961	22,907	22,736	23,458	<0.001	<0.001	0.774	5.6
7	22,502	22,886	22,824	22,627	23,455	<0.01	<0.01	0.768	5
8	22,420	22,853	22,783	22,560	23,496	>0.05	>0.05	0.762	4

Note. AIC = Akaike’s information criterion; BIC = Bayesian information criterion; SABIC = sample-size-adjusted Bayesian information criterion; AWE = approximate weight of evidence criterion; LRT = Lo–Mendell–Rubin test; AdjLRT = adjusted LRT.

**Table 3 ijerph-20-00967-t003:** Estimated means and standard errors for job demands and resources in the four profiles.

Variable	Resourceless M (SE)	Resourceful M (SE)	High Strain–Isolated M (SE)	Active Job on the Ward M (SE)	F Value	Partial η^2^
Workload	0.5 (0.1) ^a^	−0.6 (0.04) ^b^	0.6 (0.03) ^a^	−0.5 (0.03) ^b^	292.13 ***	0.40
Emotional dissonance	−0.2 (0.1) ^bc^	−0.1 (0.1) ^b^	0.4 (0.03) ^a^	−0.4 (0.04) ^c^	88.52 ***	0.17
Physical demands	−1.3 (0.1) ^d^	−0.7 (0.04) ^c^	0.2 (0.03) ^b^	0.7 (0.03) ^a^	416.29 ***	0.49
Patient demands	−1.1 (0.1) ^d^	−0.8 (0.04) ^c^	0.2 (0.03) ^b^	0.7 (0.04) ^a^	385.78 ***	0.47
Control	−1.1 (0.1) ^d^	0.6 (0.04) ^a^	−0.4 (0.03) ^c^	0.4 (0.04) ^b^	230.41 ***	0.34
Peers’ support	−1.1 (0.1) ^c^	0.6 (0.04) ^a^	−0.5 (0.03) ^b^	0.5 (0.04) ^a^	314.40 ***	0.42
Management support	−1.5 (0.1) ^d^	0.4 (0.04) ^b^	−0.4 (0.03) ^c^	0.6 (0.04) ^a^	333.34 ***	0.43

Note. SE = standard errors. Means with different superscripts are significantly different using Bonferroni post hoc test. *** *p* < 0.001.

**Table 4 ijerph-20-00967-t004:** Socio-demographic and occupational characteristics for the four latent profiles.

Variable	Resourceless	Resourceful	High Strain–Isolated	Active Job on the Ward	Test Statistic	Effect Size
*Gender* (N, %)					χ^2^_(3)_ = 8.63 *	0.08 ^a^
Male	40 (35.7)	69 (24.1)	131 (25.2)	120 (30.5)		
Female	72 (64.3)	217 (75.9)	389 (74.8)	273 (69.5)		
*Age* (N, %)					χ^2^_(6)_ = 27.17 ***	0.10 ^b^
Up to 30	4 (3.6)	13 (4.5)	28 (5.4)	34 (8.6)		
31 to 50	67 (60.4)	126 (43.9)	277 (53.4)	226 (57.4)		
More than 50	40 (36)	148 (51.6)	214 (41.2)	134 (34)		
*Type of contract* (N, %)					χ^2^_(12)_ = 17.36	0.12 ^b^
Permanent	106 (94.6)	268 (93.4)	490 (94.2)	358 (90.9)		
Fixed term	4 (3.6)	10 (3.5)	24 (4.6)	21 (5.3)		
Collaboration	0 (0)	0 (0)	1 (0.2)	4 (1)		
Temporary	0 (0)	6 (2.1)	3 (0.6)	8 (2)		
Other	2 (1.8)	3 (1)	2 (0.4)	3 (0.8)		
*Shift work* (N, %)					χ^2^_(3)_ = 115.11 ***	0.30 ^a^
Yes	77 (68.1)	100 (34.8)	350 (67.2)	285 (72.3)		
No	36 (31.9)	187 (65.2)	171 (32.8)	109 (27.7)		
*Organizational tenure in years* (M, SD)	16 (9.7)	18.9 (11.3)	16.2 (11.2)	14.8 (11.6)	*F*_(3)_ = 7.51 ***	0.02 ^c^

Note. N = number of subjects; M = mean; SD = standard deviation; ^a^ = Cramer’s V; ^b^ = Phi; ^c^ = partial eta squared; *** *p* < 0.001; * *p* < 0.05.

**Table 5 ijerph-20-00967-t005:** Bayesian evaluation of the study informative hypotheses (dependent variable = job satisfaction).

Informative Hypotheses	(In)Equality Constraints	Bayes Factor (BF)	Posterior Model Probability (PMP)
**H_0_**	µActive = µResful = µResless = µStrain	0.00	0.00
**H_1_**	µActive > µResful > µResless > µStrain	23.08	0.45
**H_2_**	µResful > µActive > µResless > µStrain	0.00	0.00
**H_3_**	µActive > µResful > µStrain > µResless	1.23	0.02
**H_4_**	µActive > µResful > µResless = µStrain	27.43	0.53
**H_5_**	µActive = µResful > µStrain = µResless	0.02	0.00
**H_6_**	µResful > µActive > µStrain > µResless	0.00	0.00
**H_7_**	µActive = µResful > µResless > µStrain	0.01	0.00
**H_8_**	µActive = µResful > µStrain > µResless	0.00	0.00
**H_9_**	µResful > µActive > µResless = µStrain	0.00	0.00
**BF_4,1_**		1.19	

Note. BF_4,1_ = informative evidence of H_4_ over H_1_. BFs and PMPs of the two most likely informative hypotheses are in bold.

**Table 6 ijerph-20-00967-t006:** Paired comparisons between latent profiles in job satisfaction.

Comparisons		Mean Difference (SE)	Cohen’s d
Resourceless	Resourceful	−0.33 (0.1)	−0.34 **
	High strain–isolated	0.16 (0.1)	0.17
	Active job on the ward	−0.66 (0.1)	−0.68 ***
Resourceful	High strain–isolated	0.50 (0.1)	0.51 ***
	Active job on the ward	−0.33 (0.1)	−0.33 ***
High strain–isolated	Active job on the ward	−0.82 (0.1)	−0.84 ***

Note. *** *p* < 0.001; ** *p* < 0.01.

## Data Availability

The datasets generated and analyzed during the current study are not publicly available. Indeed, this study uses data collected within the methodology developed by the Italian Workers’ Compensation Authority (INAIL) for the assessment and management of work-related stress factors, which are only accessible to participating organizations and to INAIL for research studies. Thus, the data are available from the corresponding author only on reasonable request.

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
