# Peer review of "Demand–Resource Profiles and Job Satisfaction in the Healthcare Sector: A Person-Centered Examination Using Bayesian Informative Hypothesis Testing"

_ijerph, 2023, doi:10.3390/ijerph20020967_

Round 1

Reviewer 1 Report

1、 In subsection 2.1firstly, a questionnaire was distributed to 6687 employeesbut a total of 1905 questionnaires were collected; of these, 1513 were healthcare providers. The questionnaire recovery rate is only about 28.5%. Will it affect the sample representativeness of this article? Secondly, more than two-thirds of participants (73.4%) were females, and 26.6% men, does the imbalance between men and women affect the accuracy of the results?

2、 In subsection 2.3, data analysis only introduces the analysis method, without elaborating the variables and research problems in this paper, so the process of data processing is unclear.

3、 A series of informative hypotheses made in Table 5 seem to be independent of other contents in the article. What is the relationship between this and the four latent profiles divided in Table 3 and Table 4?

4、 In the Abstract, the research conclusions should be briefly presented, not just the research ideas and methods. In addition, subsection 3.1 should be “Descriptive statistics results”.

5、 The sources of employee well-being are diversified. Is it accurate to measure well-being only by job satisfaction in this paper?

Author Response

Dear Editor and manuscript reviewers,

Thank you very much for your valuable comments regarding our manuscript (ijerph- 2125333) submitted for consideration in International Journal of Environmental Research and Public Health.

We greatly appreciated your constructive suggestions. They served us to further strengthen our arguments and improve the manuscript’s clarity. Please find below a detailed summary of our responses to each concern and a description of the resulting changes made to address these concerns. All these edits are highlighted in yellow in the revised manuscript.

We thank you again for your helpful feedback and look forward to your evaluation of the revised manuscript. 

Sincerely,

The Authors.

Please find below detailed answers to your comments. 

  1. In subsection 2.1,firstly, a questionnaire was distributed to 6687 employees, but a total of 1905 questionnaires were collected; of these, 1513 were healthcare providers. The questionnaire recovery rate is only about 28.5%. Will it affect the sample representativeness of this article? Secondly, more than two-thirds of participants (73.4%) were females, and 26.6% men, does the imbalance between men and women affect the accuracy of the results?

Response: We have provided a more in-depth explanation about the sample and procedures of the study. Factors determining a low response rate are different. This may depend on the procedure of data collection, the type of sample or a lack of trust in surveys. As regard to a low response rate into a hospital using an online survey, two main reasons should be considered: 1) these are context where surveys are really frequent and people might feel overwhelmed by questionnaires; 2) most of the healthcare providers have a more limited access to a computer during their work than other kind of professions. 

Nevertheless, even if the response rate remains informative, we do not think this necessarily have an impact on the study validity. As regard to this, in their Editorial on the Australian and New Zealand Journal of Public Health, Morton and colleagues (2012) discussed about how much has changed the response rate over the years, and which kind of response rate could be considered acceptable in 21st Century. Some evidence is reported into the editorial about the lack of clear direct correlation between response rate and validity. Thus, low response rate should not be necessary related to a low validity, this simply indicates a potentially greater risk.

Regarding gender, the proportion of women in our sample fully reflects the quota of the reference working population. In the last decades, Italy assisted to a phenomenon called “feminization” of the healthcare providers, that led healthcare becoming a high gendered sector. There are over 70% women working in the National Health Service, and this proportion is mirrored into our sample.

Moreover, as regard to the existence of gender differences in the perceptions of stressors, we found mixed findings from the literature and research findings seems to be highly inconsistent (Gyllensten et al., 2005). Thus, we cannot assert that this sample proportions affect the accuracy of the results.

References:

Gyllensten, K., & Palmer, S. (2005). The role of gender in workplace stress: a critical literature review. Health Education Journal64(3), 271–288. https://doi.org/10.1177/001789690506400307

Morton, S. M., Bandara, D. K., Robinson, E. M., & Carr, P. E. (2012). In the 21st Century, what is an acceptable response rate? Australian and New Zealand journal of public health36(2), 106–108. https://doi.org/10.1111/j.1753-6405.2012.00854.x

  1. In subsection 2.3, data analysis only introduces the analysis method, without elaborating the variables and research problems in this paper, so the process of data processing is unclear.

Response: Thank you for the suggestion. We have improved the methods section and clarified the analytical strategies we employed to answer our research questions. Moreover, we described better which variables were considered for each step of the analysis.

  1. A series of informative hypotheses made in Table 5 seem to be independent of other contents in the article. What is the relationship between this and the four latent profiles divided in Table 3 and Table 4?

Response: All the informative hypotheses listed in Table 5 have been now described and linked to previous literature in the subsection “3.5. Association between JD-R profiles and job satisfaction”.

  1. In the Abstract, the research conclusions should be briefly presented, not just the research ideas and methods. In addition, subsection 3.1 should be “Descriptive statistics results”.

Response: We have improved the abstract, which now briefly includes the research conclusions. Moreover, as suggested, we renamed subsection 3.1 in “Descriptive statistics results”.

  1. The sources of employee well-being are diversified. Is it accurate to measure well-being only by job satisfaction in this paper?

Response: This is a very good point, thank you for having raised this important issue. Unfortunately, as the questionnaire was already very long, we had to make choices on which indicators of well-being to include in order to prevent participants from feeling overwhelmed. We decided to focus on job satisfaction for different reasons. First, a meta-analysis of Faragher and colleagues demonstrated that this construct is strongly linked with employee global health (Faragher et al., 2007). Second, job satisfaction may have important consequences also on an organization-level, being linked with high performance and productivity, reduced absenteeism, lower turnover intention and less counterproductive work behaviors (e.g., Wright & Cropanzano, 2000). Third, we believe that job satisfaction is really critical for the healthcare sector, as prior studies demonstrated that it is associated with higher quality of care, effectiveness, commitment to work, patient satisfaction, lower rates of medical errors and higher patient safety (Liu et al., 2016; Bhatnagar & Srivastava, 2012). However, we recognize that future studies could benefice from the inclusion of additional pathogenic (e.g., burnout, psychological distress) or salutogenic outcomes (e.g., work engagement, commitment) to investigate if similar results would be obtained.

In light of your question, we added additional information on job satisfaction in the introduction and in the paragraph on the study limitations and future suggestions.

References:

Bhatnagar, K., & Srivastava, K. (2012). Job satisfaction in health-care organizations. Industrial Psychiatry Journal21(1), 75-78. https://doi.org/10.4103/0972-6748.110959

Faragher, E. B., Cass, M., & Cooper, C. L. (2005). The relationship between job satisfaction and health: a meta-analysis. Occupational and Environmental Medicine62(2), 105-112. https://doi.org/10.1136/oem.2002.006734

Liu, Y., Aungsuroch, Y., & Yunibhand, J. (2015). Job satisfaction in nursing: A concept analysis study. International Nursing Review, 63(1), 84-91. https://doi.org/10.1111/inr.12215

Wright, T. A., & Cropanzano, R. (2000). Psychological well-being and job satisfaction as predictors of job performance. Journal of Occupational Health Psychology, 5(1), 84–94. https://doi.org/10.1037/1076-8998.5.1.84

Reviewer 2 Report

1.       Abstract

a.       Add research gap

b.       Add research novelty

2.       Introduction

a.       Give the newest references

b.       Add research gap clearly

c.       Add research novelty clearly

d.       Explanations between variables do not connect with each other

e.       The conceptual framework does not consistent with the title of this research  

3.       Research Methods

a.       Good explanations

4.       Results and Discussions

a.       The explanation of each hypothesis is lacking in detail

b.       Give the newest references to support the discussions

5.       Conclusion

a.       Good explanations 

Author Response

Dear Editor and manuscript reviewers,

Thank you very much for your valuable comments regarding our manuscript (ijerph- 2125333) submitted for consideration in International Journal of Environmental Research and Public Health.

We greatly appreciated your constructive suggestions. They served us to further strengthen our arguments and improve the manuscript’s clarity. Please find below a detailed summary of our responses to each concern and a description of the resulting changes made to address these concerns. All these edits are highlighted in yellow in the revised manuscript.

We thank you again for your helpful feedback and look forward to your evaluation of the revised manuscript. 

Sincerely,

The Authors.

Please find below detailed answers to your comments. 

  1. Abstract
    1. Add research gap
    2. Add research novelty

Response: The abstract has been improved. We have briefly described research gaps, research novelty and main conclusions.

  1. Introduction
  2. Give the newest references

Response: We have improved the paragraphs and subsections of the introduction by citing up-to-date research, especially related to the healthcare sector and the influence that the pandemic has had on this work environment during the last few years.

  1. Add research gap clearly

Response: We have described better which are in our opinions the main limitations of previous research on this topic. In specific, we have underlined that prior studies mainly focused on global psychosocial risks such as workload, control and support, although it is fundamental to include both “generic” and “occupation-specific” work characteristics in any research designed to investigate occupational health (i.e., Brough & Biggs, 2015). Moreover, we have described in greater detail the differences among “variable-centered” and “person-centered” approaches, underlining why variable-centered methodologies are not well-suited to investigate the complex interactions between different variables, such as many demands and resources.

References:

Brough, P., & Biggs, A. (2015). Job demands × job control interaction effects: do occupation-specific job demands increase their occurrence? Stress and health: journal of the International Society for the Investigation of Stress31(2), 138–149. https://doi.org/10.1002/smi.2537

  1. Add research novelty clearly

Response: We have underlined that very few studies have investigated our research questions by focusing on both generic and contextual risk factors and using a person-centered methodology. Moreover, to the best of our knowledge, the current research is the first which employed a Bayesian informative hypothesis testing approach to investigate the association among different JD-R profiles and employee well-being. The main strengths of this approach have been now detailed in the subsection “1.4. Association between JD-R profiles and employee well-being”

  1. Explanations between variables do not connect with each other

Response: We have improved the subsection “1.1. Background: the JD-R model and job satisfaction”. In specific, we have described better the JD-R model and its two processes (health impairment and motivational processes). Moreover, we have detailed the potential influence that generic and specific work characteristics may have on well-being indicators such as job satisfaction in order to better link the variables.

  1. The conceptual framework does not consistent with the title of this research

Response: The title has been slightly modified in order to underline that we used the JD-R model as our theoretical framework.

  • Research Methods
  1. Good explanations

Response: Thank you.

  1. Results and Discussions
  2. a) The explanation of each hypothesis is lacking in detail

Response: Thank you for raising this point. All the informative hypotheses listed have been now described in detail in the subsection “3.5. Association between JD-R profiles and job satisfaction”.

  1. Give the newest references to support the discussions

Response: As in the introduction, we have now cited up-to-date references especially related to the healthcare sector and the influence that the pandemic has had on this work environment during the last few years (i.e., Abraham et al., 2021; Franklin & Gkiouleka, 2021; ILO, 2020). Moreover, we have added additional, recent references to strengthen our methodological framework (e.g., Spurk et al., 2020; Vignoli et al., 2017; Kluytmans et al., 2012).

References:

Abraham, A., Chaabna, K., Doraiswamy, S. et al. Depression among healthcare workers in the Eastern Mediterranean Region: a systematic review and meta-analysis. Hum Resour Health 19, 81 (2021). https://doi.org/10.1186/s12960-021-00628-6

Franklin, P., & Gkiouleka, A. (2021). A Scoping Review of Psychosocial Risks to Health Workers during the Covid-19 Pandemic. International journal of environmental research and public health18(5), 2453. https://doi.org/10.3390/ijerph18052453

ILO (2020). Managing work-related psychosocial risks during the COVID-19 pandemic. International Labour Office

Kluytmans, A., Schoot, R., Mulder, J., & Hoijtink, H. (2012). Illustrating bayesian evaluation of informative hypotheses for regression models. Frontiers in Psychology, 3(2) https://doi.org/10.3389/fpsyg.2012.00002

Spurk, D., Hirschi, A., Wang, M., Valero, D., & Kauffeld, S. (2020). Latent profile analysis: A review and “how to” guide of its application within vocational behavior research. Journal of Vocational Behavior, 120, Article 103445. https://doi.org/10.1016/j.jvb.2020.103445

Vignoli, M., Nielsen, K., Guglielmi, D., Tabanelli, M. C., & Violante, F. S. (2017). The Importance of Context in Screening in Occupational Health Interventions in Organizations: A Mixed Methods Study. Frontiers in psychology8, 1347. https://doi.org/10.3389/fpsyg.2017.01347

  1. Conclusions
  2. a) Good explanations

Response: Thank you.

Reviewer 3 Report

Health workers’ Job satisfaction issue is always a popular research topic that determines the working outcomes and quality of care. Therefore there are many available publications around the world. So this research topic is not new, but the authors draw some interesting discussions. Several suggestions or questions as below.

1. Since it is a self-reported survey, how the research team invites the hospital staff to join this survey? and two hospitals' basic information are similar? how to choose these two? any geomorphic/patient background similarity? both are public or private hospitals?

2. Many different measurements were involved in this research, how to define ‘ contextual work characteristics’ and whether the participant could understand it? For avoiding misunderstanding wrong expectations? any education level differences?

3. The results showed ‘resourceful and active job on the ward profile were the most satisfied’ the findings matched previous many published papers. But how to apply it in the real world needs more solid and balance suggestions.

4. Did not see any specific analysis to separate the administration workers and professional caregivers, please explain more about this

Author Response

Dear Editor and manuscript reviewers,

Thank you very much for your valuable comments regarding our manuscript (ijerph- 2125333) submitted for consideration in International Journal of Environmental Research and Public Health.

We greatly appreciated your constructive suggestions. They served us to further strengthen our arguments and improve the manuscript’s clarity. Please find below a detailed summary of our responses to each concern and a description of the resulting changes made to address these concerns. All these edits are highlighted in yellow in the revised manuscript.

We thank you again for your helpful feedback and look forward to your evaluation of the revised manuscript. 

Sincerely,

The Authors.

Please find detailed point by point answers to your comments as follows.

Health workers’ Job satisfaction issue is always a popular research topic that determines the working outcomes and quality of care. Therefore there are many available publications around the world. So this research topic is not new, but the authors draw some interesting discussions. Several suggestions or questions as below.

  • Since it is a self-reported survey, how the research team invites the hospital staff to join this survey? and two hospitals' basic information are similar? how to choose these two? any geomorphic/patient background similarity? both are public or private hospitals?

Response: We realized we did not include a fully explanation about procedures of sampling and data collection. We have added more information into the methods thanks to your comment. Particularly, as regard to the hospitals, we would explain that these are both public hospitals, with really similar characteristics in terms of size, missions, departments and units, and resources. INAIL included such hospitals in the study since both must conduct a psychosocial risks assessment and management and these had the common need of evaluating specific risk factors of this sector.

  • Many different measurements were involved in this research, how to define ‘ contextual work characteristics’ and whether the participant could understand it? For avoiding misunderstanding wrong expectations? any education level differences?

Response: We have added additional information on what we intended for “contextual work characteristics”, and why it is important to include both generic and occupation-specific variables in any assessment of the psychosocial work environment.

As regards to the involvement of the participants, we have included additional information on the procedure and our general aims. These were clearly described by INAIL to the hospitals and the employees involved in the project before the administration of the questionnaire. Unfortunately, we did not include an item on the education level of the participants; thus, we are not able to provide additional analyses on this aspect.

  • The results showed ‘resourceful and active job on the ward profile were the most satisfied’ the findings matched previous many published papers. But how to apply it in the real world needs more solid and balance suggestions.

Response: The subsection “4.1. Implications for practice in the COVID-19 era” has been enriched with new suggestions and practical examples for human resources specialists and managers.

  • Did not see any specific analysis to separate the administration workers and professional caregivers, please explain more about this

Response: We have provided a more in-depth explanation about the sample and procedures of the study. To focus on similar roles and capture as most as possible shared job peculiarities, in this study we decided to include only the healthcare providers, namely 1,513 participants. We believe difference between administrative and healthcare providers were not in line with our purpose of focusing on different specific demand and resource profiles in the healthcare sector.

Round 2

Reviewer 1 Report

Accept in present form